# Real-Time Model-Free Minimum-Seeking Autotuning Method for Unmanned Aerial Vehicle Controllers Based on Fibonacci-Search Algorithm

**DOI:** 10.3390/s19020312

**Published:** 2019-01-14

**Authors:** Wojciech Giernacki, Dariusz Horla, Tomáš Báča, Martin Saska

**Affiliations:** 1Institute of Control, Robotics and Information Engineering, Poznan University of Technology, Piotrowo 3a, 60-965 Poznan, Poland; dariusz.horla@put.poznan.pl; 2Department of Cybernetics, Faculty of Electrical Engineering, Czech Technical University in Prague, Technicka 2, 166 27 Prague 6, Czech Republic; tomas.baca@fel.cvut.cz (T.B.); martin.saska@fel.cvut.cz (M.S.)

**Keywords:** UAV, auto-tuning, extremum-seeking control, iterative learning, optimization

## Abstract

The paper presents a novel autotuning approach for finding locally-best parameters of controllers on board of unmanned aerial vehicles (UAVs). The controller tuning is performed fully autonomously during flight on the basis of predefined ranges of controller parameters. Required controller properties may be simply interpreted by a cost function, which is involved in the optimization process. For example, the sum of absolute values of the tracking error samples or performance indices, including weighed functions of control signal samples, can be penalized to achieve very precise position control, if required. The proposed method relies on an optimization procedure using Fibonacci-search technique fitted into bootstrap sequences, enabling one to obtain a global minimizer for a unimodal cost function. The approach is characterized by low computational complexity and does not require any UAV dynamics model (just periodical measurements from basic onboard sensors) to obtain proper tuning of a controller. In addition to the theoretical background of the method, an experimental verification in real-world outdoor conditions is provided. The experiments have demonstrated a high robustness of the method to in-environment disturbances, such as wind, and its easy deployability.

## 1. Introduction

Controller tuning precision significantly influences in-flight properties of any architecture used to control multirotor unmanned aerial vehicles (UAVs). Effective and precise tracking of a desired state during flight, precise positioning, and autonomous landing can be ensured by nonlinear controllers with complicated structure and adaptation mechanisms [1]. However, they usually require knowledge about UAV dynamics, which results in a large computational burden when calculating a control law, made additionally more severe, by the introduction of optimality criteria. Alternatively, fixed-parameter controllers of PD (proportional-derivative) or PID (proportional-integral-derivative) types are nowadays widely used [2,3,4,5,6].

When these controllers are tuned appropriately, they can ensure sufficient performance in most of the required UAV applications, including demanding active interaction with the environment. Furthermore, they neither require the knowledge of the analytical model of UAVs nor they require performing any online estimation of the parameters. The introduction of an appropriate optimality criterion in the form of a cost function streamlines the tuning procedure and synthesis of a controller. In the problem considered in this paper, it is assumed, in addition, that the tuning procedure of a controller (or performing fine-tuning for rough initial estimates of controller parameters) is performed in real time during flight of a UAV. This is especially useful in applications such as transporting load by a single flying unit [7,8] or by multiple cooperating UAVs [9] that require even better precision of trajectory tracking due to the coupling between the robots, e.g., in grasping tasks (see Figure 1). When the load changes, it is important to tune controller parameters to the new dynamics of the system. In the presented approach, it is also possible in the ideal case of no model-UAV mismatch (perfect modeling), to obtain an optimal controller parameters for various flight scenarios, e.g., in acrobatic maneuvres (with dynamic acceleration phase) or maximization of flight time by avoiding sudden moves using smooth and feasible reference trajectories.

The work presented in this paper has been motivated by the Mohamed Bin Zayed International Robotics Challenge 2017 (MBZIRC) in Abu Dhabi, where a precise altitude controller has shown to be one of the key aspects of proper system behavior in both tasks designed for UAVs. In the first challenge, landing on a fast moving vehicle, fast descent and precise acquirement of the required altitude are obviously crucial for a safe approach and soft touch-down onto the helipad. Overshooting the set point can result in a too fast descent and consequently failing to attach the device due to a too aggressive impact. In the third challenge, a treasure hunt, in which a set of static and dynamic objects has to be located, tracked, picked and placed, a precisely tuned controller is even more important to achieve the desired performance.

The necessity of frequent controller tuning has been highlighted in the MBZIRC competition, which required a fleet of multiple UAVs with different sensory equipment and actuators, but its importance seems as big in any deployment of UAV systems. Very precise manual tuning of one set of controller parameters takes days of flying outdoor in different environment conditions. Moreover, each change of the platform (mainly those that influence UAV weight) requires obtaining a new set of parameters and therefore additional time-consuming on flight parameter analyses is needed. Therefore, the aim of this paper is to design an advanced method for parameter tuning that achieves even better results than manual tuning in a fraction of time required by current approaches.

In the paper, a novel method for real-time model-free minimum-seeking tuning of controllers is presented (FGT, Fibonacci-search Gain Tuning), including verification of its applicability in a sample application of in-flight tuning of an altitude controller of the PD type in a hexacopter UAV. This method is universal to such an extent that it can be as well adapted for tuning of any other controller (in general, of different types too) used in multirotor and fixed-wing UAVs. The effective tuning of a controller is ensured by the Fibonacci-search method, which rapidly reduces the search space, which is especially important in the field of UAVs from practical reasons, i.e., a limited flight duration, in which both the voltage of the batteries and thrust forces of flying robots decrease in time [10]. The proposed approach considers possible constraints imposed on a control signal as well as the measurement noise and external disturbances the effect of which is compensated using a low-pass filter. The output signal of a closed-loop system is assumed to originate from a single or multiple sources (data fusion), e.g., various on-board sensors, GPS or any external motion capture system.

## 2. State of the Art


PD and PID type controllers with fixed parameters mentioned in previous subsection are widely used in UAVs and are a good starting point to characterize the current state of the art considering their tuning and techniques to enhance their performance. From a historical point of view, early methods, such as [11,12], enable one to obtain controller parameters for plants in various structures of different nature and dynamics, usually leading to stable closed-loop systems. However, the obtained parameters are not optimal since there is no optimality criterion introduced. Over the years, multiple model-based or model-free approaches have appeared, and an in-depth review of 51 practice-oriented methods for a PID controller design for specified performance criteria can be found in [13]. The authors would like to focus readers’ attention on the third edition of the book [14] in which an exhaustive review of tuning methods used between 1935 and 2008 for PI and PID controllers has been made along with the proceedings of the cyclic conference *IFAC Conference on Advances in Proportional-Integral-Derivative Control*.

In parallel to the increase in computational power of computers, batch optimization, soft-computing, and methods of optimal controller tuning have appeared, which are still developing up to this date. Such methods require cost functions to be defined, expressing designer demands with the mathematical language, imposed on control systems, such as stability, robustness against disturbances, predefined dynamics of selected signals, either in steady or transient state, and overshoot requirements or expected steady-state properties. In the simplest and widely-encountered applications, the cost functions have been constructed using an absolute or squared tracking error or control signal, leading to integral performance indices, such as IAE, ISE, ITAE, ITSE, etc. The problem of parameter tuning, however, does not require in general any knowledge concerning plant dynamics, which can be identified as the greatest advantage of this approach. Later in the text, the focus turns to model-free tuning procedures. Optimization methods based on fuzzy logic [15,16] or neural networks [17,18] used to implement and tune control algorithms have been treated as out of scope in this paper due to the reasons analogous to these pointed out in [19]. In the latter reference, a pure definition of a model-free approach is stated as follows: *"However, we question most of these claims on the basis of the hidden or implicit system modeling required. (…) For example, fuzzy controllers are frequently claimed as model-free; nevertheless, in all fuzzy controllers there is a requirement for a rules base (or associative memory matrix) that describes the dynamics of the system in a linguistic-type fashion."* The authors decided to characterize current state of the art and give the background to the proposed original approach in the context of two main optimization strands: bio-inspired metaheuristics and global deterministic optimization methods.

In the past few years, there has been a noticeable increase in interest in bio-inspired methods in various applications [20]. These metaheuristic methods independently of their major drawbacks (no guarantee to find the global optimum of the cost function, necessary prototyping of solutions using computer simulations, and—usually—a long calculation time) are used nowadays in both model-based and model-free controller tuning procedures. As an example, Ref. [21] proposed fixed-gain controllers in PID type for a four-rotor coaxial mathematical model with multiple additional effects taken into account, such as drag or friction forces in a PSO (Particle Swarm Optimization) framework. The idea and the principles of this metaheuristics, as well as others, such as Ant Colony Optimization, Artificial Bee Colony, Glowworm Swarm Optimization, Bacteria Foraging Optimization, Bat-inspired algorithm, have been presented in [22], which, in a wide context, presents computational bio-inspired algorithms applied in the field of UAVs.

In this paper, the authors propose an alternative approach to the methods listed in the above paragraph to perform automated tuning of controllers, using deterministic optimization methods. It is based on the assumption that the obtained solution for a specific type of a cost function is a global minimizer with prescribed tolerance, calculated in a defined time regime. From multiple available approaches to tuning problems, easily-implementable iterative optimization methods have been selected and expected to allow either tuning or fine-tuning of controllers during flight of UAVs. Their computational complexity and conceptual simplicity is an advantage in comparison with multi-agent based optimization techniques, where time necessary to conduct all computations is proportional to the number of agents. The approach in this paper is inspired by a classic Iterative Feedback Tuning method [23,24,25] in which optimal controller parameters are sought in reference to some cost function calculated using a current output signal of a closed-loop system. The references mention the latter method among such approaches as Fictitious Reference Iterative Tuning [26,27,28], Iterative Learning Approach [29,30] or Virtual Reference Feedback Tuning [31]. The latter methods enable direct tuning based on experimental input-output data. Direct methods can be divided into classes such as zero- or first-order optimization techniques. The first class uses only zero-order interval arithmetic, as in the bisection, golden-search or Fibonacci-search method, and the second one uses interval gradients or interval slopes, as in IFT or ILA, Steepest Descent Method, Reinforcement Learning [32] or methods proposed in [30,33].

The literature terms zero-order optimization methods alternatively as derivative-free optimization methods.

The analysis of the literature indicates that there are just few papers so far which discuss the use of golden-search and Fibonacci-search methods in the context of derivative-free optimization used for control purposes. This might be due to the reason that these methods are considered to be slower in comparison with gradient-based approaches [34]. In addition, as stated in [35], *"there has recently been renewed interest in optimization problems with only functional (zero-order) information available rather than first-order gradient information in optimization, machine learning, and statistics"*, where the reference to [36,37,38] is given.

Ref. [39] presents a novel strategy of maximum power point tracking for photovoltaic power generation systems based on the Fibonacci-search technique. The aim of the paper has been to realize a simple control system to track the real maximum power point even under non-uniform or for rapidly changing insolation conditions. In the article [40], a practical PI/PID controller tuning method for integrating processes with dead time and inverse response based on a model has been proposed. Optimum tuning of the parameter for disturbance rejection based on the model and the minimum of the IAE criterion has been successfully obtained via adaptation of the golden-search technique. However, the approach presented in the current paper aims to result in obtaining an iterative tuning scheme of controller parameters independently of the availability of the model, as shown in [41], where, to the best knowledge of the authors, there is only a single reference available in the literature referring to the concept of using Fibonacci-search method in iterative model-based PID controller tuning. Ref. [41] unfortunately delivers only a blurry and general procedure for possible search in parameter space stating that the reciprocated Fibonacci sequence which has been used to adjust the controller parameters has been chosen arbitrarily because it is a convenient, convergent sequence. The paper does not present in full the potential and possible advantages of the use of this algorithm in controller tuning procedures, which is the basic reason for writing this paper and formulating further research.

In the presented approach, the authors went beyond the topics of the above-mentioned papers in several aspects. Ordering the novel ideas presented in the paper: the added value to this work is the iterative tuning procedure, seeking for the minimum of a function, of a time-invariant controller with small number of parameters, i.e., 1–3, for either slow or fast processes, where availability of a plant model is not required. The controller is assumed to have a constant structure, with its parameters changed when the tuning algorithm runs. It results in a modified behaviour of the closed-loop system, though selecting the gains in an orderly manner, i.e., according to the presented algorithm, a similar search procedure is obtained as one of a human, who seeks for the best gains. The considered tuning procedure is, however, fully automated. As it does not need indeed in the ideal case any model (see comments to Remark 4—the optimality will be henceforth used to present optimality of a function, not control), the final decision in the worst case is locally optimal, in the best case—globally optimal, depending on the characteristics of the environment and the proper method chosen. The model free feature of the presented tuning method is, actually, its main novelty, and the greatest difference from the existing tuning methods. Knowing safe ranges of parameters (ensuring stability), this is simply a plug-and-tune method. The method is designed to enable tuning with averaging the already-considered pairs of gains along multiple iterations, which results in diminishing the outlier impact on the final tuning result, or to include a low-pass filter to the signal which is used to build the cost function. Both of the above modifications of the method allow one to fit it to the point of its application. Sample tuning of a PD controller has been theoretically described and verified by means of experiments in controlling altitude in flight conditions of a hexacopter UAV, where the Fibonacci-search method is used together with a bootstrap approach in obtaining optimal controller parameters in a duration time-limited experiment. In addition, three modifications of the method have also been proposed and verified, aiming at increasing both the speed and accuracy of the procedure of obtaining controller parameters. During implementation, a low-pass filtering has been used and tested to verify robustness of the procedure against measurement noise both in simulations and during experiments using a hexacopter UAV in real-world conditions.

The paper is structured as follows: a brief statement of the considered practical tuning problems regarding UAVs controllers is summarized in Section 3. In Section 4, the Fibonacci-search Gain Tuning method is described. Subsequent subsections present a full description of the theoretical basis necessary to understand and implement the method both in flight simulators and systems embedded in UAVs. The testing platform is presented in Section 5. Numerical and hardware experiments are presented in Section 6 followed by the authors’ conclusions in Section 7. Finally, the Appendix gives a detailed message printed out of the console obtained during the experiment of minimum-seeking tuning of controller parameters described in the paper, concerning real-world hexacopter UAV robot using the FGT method.

## 3. UAV Applications—Practical Problems

The authors of the paper are mainly interested in applicability of the FGT method in the field of multirotor UAVs operating in various tasks and on different missions [42,43,44,45,46,47], including multi-UAV applications, where the highest level of reliability, safety and maneuvering precision is required. To shed a light on the scope of possible applications, a general idea has been presented in Figure 1 comprising a set of current applications. Each of the applications requires the use of effective position and orientation controllers. Furthermore, prior experience of the authors gained from the MBZIRC 2017 (Figure 1b,c) stresses the need to be able to stabilize the robot at a given altitude and to control its autonomous vertical maneuvers and landing, especially in conditions with outdoor-like disturbances, at the highest level of precision. This is the core problem that has been encountered in the field of robotics in UAVs for years, both in RC-controlled flights of simple, low-cost constructions as well as in advanced autonomous systems. An application of fixed-parameter controllers, such as PDs or PIDs, is very often connected to tuning which mirrors a compromise to find parameters for drastically different flying conditions accompanied by dynamically changing altitude during flight. For example, one type of a controller response is needed during the take-off phase in acrobatic contests, another one for precise landing and yet another one when recording smooth video footage or during manipulating with objects lifted by the UAV. This explains why it is desired to ensure the possibility to tune or to change controller parameters of UAV during flight in as short time as possible, and for the problems connected with prototyping new machines—their fine-tuning.

The problem considered in the paper is confined to finding the locally-best parameters of a PD controller based on a given reference altitude primitive and measured altitude in a predefined accumulative cost function which is calculated in flight conditions for a hexacopter UAV with given safe ranges of controller parameters. This enables the procedure to find controller parameters, taking preferred dynamics and initial controller parameters into account, for the given tolerance of the solution. The tuning method is used in a control system with a convex performance index in a form of a sum of absolute values of the tracking error. In the usual case, and when averaged over a longer time horizon, the majority of controller design tasks present their convex nature. This is the main point of applicability of the presented method. However, when run a single time in a highly-disturbed environment, the method will definitely result in optimal, though locally optimal solution. As the presented algorithm revolves around certain points in a search space, cropping the ranges from two opposite sides alternating the new ending point, a single missed decision, when impeded by disturbances, is compensated by the decisions to follow, which will be presented in due course of the paper.

## 4. Fibonacci-Search Gain Tuning

### 4.1. Preliminaries and Notation

The optimization procedure is based on the following assumptions:the model/mathematical description of the dynamical system is unknown/imprecise or does not simply allow tuning the controller parameters by analytical calculations;there is no formula describing the cost function based on which it can be verified whether the solution to the problem, i.e., controller parameters, is optimal;the value of the cost function is based on observing a performance index in a given time horizon, and can be obtained repeatedly by performing consecutive experiments during the same flight;the optimal solution (minimizer) in the form of optimal parameters of a controller is obtained in an iterative manner;a zero-order iterative algorithm (a branch and bound algorithm) has been chosen to find the solution.

The notation used in the paper to generally describe the FGT algorithm, is summarized in Table 1.

### 4.2. The Idea behind Zero-Order Iterative Methods for Finding Minimum Points

The following two definitions are considered here.

**Definition** **1.**
*A function f(x) is unimodal if for a value m, it is monotonically decreasing for x≤m and monotonically increasing for x≥m. In that case, the minimum value of f(x) is f(m) and are no other local minima exist.*


**Definition** **2.**
*The presented method applies to finding the minimum of a function f:R→R expected to lie in the range $\left[x^{\left(0^{-}\right)},\;x^{\left(0^{+}\right)}\right],$ where $x^{\left(0^{+}\right)}>x^{\left(0^{-}\right)}$, f is expected to be a unimodal function, and the argument x of a function f can be, e.g., a parameter of a controller.*


If the conditions from the above definition are violated, and the function has multiple minima, the method will find a local minimum of a multimodal function. In order to obtain the global minimum, the method should be initialized from different starting points, with cost functions compared between local minima. On the basis of the performed experiments, and observations of their results from several runs of the method, it is possible. Please refer to the case presented in Appendix B, where, in the case of a two-parameter controller with fixed parameters, the method has been initialized from various starting points (Par2). The same can be done for Par1, and on the basis of expert knowledge, by changing a range for Par1 (see Par(−) and Par(+)). It is recommended to repeat the tuning procedure an arbitrary number of times to make the result closer to the global one.

**Theorem** **1.**
*Let a unimodal function f be given (see Figure 2) and considered in the following range of its arguments: $\left[x^{\left(0^{-}\right)},\;x^{\left(0^{+}\right)}\right]$. The information concerning current value of f can be obtained for any value of its argument with no use/no information concerning a possible reduction of the considered range, where the minimum lies.*


**Proof.** In the procedure, two intermediate points to evaluate *f* are taken to symmetrically reduce the range of possible arguments, i.e.,
(1)$$x^{\left(1^{-}\right)}-x^{\left(0^{-}\right)}=x^{\left(1^{+}\right)}-x^{\left(0^{+}\right)}=\rho \left(x^{\left(0^{+}\right)}-x^{\left(0^{-}\right)}\right),$$
where a reduction factor $\rho <\frac{1}{2}$, as in Figure 3.When the values of *f* at the two intermediate points are known, the following cases apply:
$f\left(x^{\left(1^{-}\right)}\right)<f\left(x^{\left(1^{+}\right)}\right)$→ the minimum is in the range $\left[x^{\left(0^{-}\right)},\;x^{\left(1^{+}\right)}\right]$ (see Figure 4);$f\left(x^{\left(1^{-}\right)}\right)\geq f\left(x^{\left(1^{+}\right)}\right)$→ the minimum is in the range $\left[x^{\left(1^{-}\right)},\;x^{\left(0^{+}\right)}\right]$. ☐

**Proposition** **1.**
*In the second iteration, two intermediate points are chosen, i.e., $x^{\left(2^{-}\right)}$ and $x^{\left(2^{+}\right)}$, function values are evaluated again, and the algorithm runs in a loop on the basis of the branch-and-bound procedure. The function f does not have to be neither differentiable nor continuous in the range $\left[x^{\left(0^{-}\right)},\;x^{\left(0^{+}\right)}\right]$, but can be simply obtained, e.g., by measuring signals and incrementing some performance index.*


**Proposition** **2.**
*For the initial range $x\in D^{\left(0\right)}=\left[x^{\left(0^{-}\right)},\;x^{\left(0^{+}\right)}\right]$, the branch-and-bound algorithm is as follows:*

*evaluate the minimal number of iterations N for which the difference between true minimum $x^{*}$ and iterative solution ${\hat{x}}^{*}$ (it is assumed that it is in the middle of the range $D^{\left(N\right)}$) does not exceed the prescribed relative accuracy ϵ, where*
(2)$$|x^{*}-{\hat{x}}^{*}|\leq \epsilon \left(x^{\left(0^{+}\right)}-x^{\left(0^{-}\right)}\right)\;,$$

*for k=1,…,N:*
*(1)* 
*pick two intermediate points ${\hat{x}}^{\left(k^{-}\right)}$, ${\hat{x}}^{\left(k^{+}\right)}$ (${\hat{x}}^{\left(k^{-}\right)}<{\hat{x}}^{\left(k^{+}\right)}$, $\left\{{\hat{x}}^{\left(k^{-}\right)},\;{\hat{x}}^{\left(k^{+}\right)}\right\}\in D^{\left(k-1\right)}$) from the current range $D^{\left(k-1\right)}$;*
*(2)* 
*obtain the new range $D^{\left(k\right)}$ selecting its limits as:*
*(a)* 
*for $f\left({\hat{x}}^{\left(k^{-}\right)}\right)<f\left({\hat{x}}^{\left(k^{+}\right)}\right)$, $x^{\left(k+1\right)}\in D^{\left(k\right)}=\left[x^{\left(k-1^{-}\right)},\;{\hat{x}}^{\left(k^{+}\right)}\right]$;*
*(b)* 
*for $f\left({\hat{x}}^{\left(k^{-}\right)}\right)\geq f\left({\hat{x}}^{\left(k^{+}\right)}\right)$, $x^{\left(k+1\right)}\in D^{\left(k\right)}=\left[{\hat{x}}^{\left(k^{-}\right)},\;x^{\left(k-1^{+}\right)}\right]$;*

*(3)* 
*put k:=k+1;*


*assume that ${\hat{x}}^{*}=\frac{1}{2}\;\left(x^{\left(N^{+}\right)}+x^{\left(N^{-}\right)}\right)$ is the solution to the problem.*



### 4.3. Fibonacci-Search Method Concept

**Assumption** **A1.**
*Let it hold that in the current range $D^{\left(0\right)}$$f\left(x^{\left(0^{-}\right)}\right)<f\left(x^{\left(0^{+}\right)}\right)$; then, $x^{*}\in D^{\left(1\right)}=\left[x^{\left(0^{-}\right)},\;x^{\left(1^{+}\right)}\right]$.*


**Proposition** **3.**
*Since the point $x^{\left(1^{-}\right)}$ is in the new range $D^{\left(1\right)}$, the value of $f(x^{\left(1^{-}\right)})$ is already known from the previous iteration (a potential reduction of the computational burden of the method), then $x^{\left(1^{-}\right)}$ should be chosen to be at $x^{\left(2^{+}\right)}$. In such a case, it is necessary to evaluate f at a single point only, i.e., $x^{\left(2^{-}\right)}$.*


This can be achieved by a special choice of $\rho_{k}$ and $\rho_{k+1}$ between iterations, which is shown in Figure 5 and Figure 6 and explained in the further parts of the text.

**Definition** **3.**
*Let us choose the reduction factor which at k-th iteration is $\rho_{k}$, and in the next one, becomes $\rho_{k+1}$, with $0\leq \rho_{k}\leq \frac{1}{2}$.*

*According to the above,*
(3)$$\rho_{k+1}\left(1-\rho_{k}\right)=1-2\rho_{k}\;,$$
*and*
(4)$$\rho_{k+1}=1-\frac{\rho_{k}}{1-\rho_{k}}\;.$$


**Remark** **1.**
*There are many sequences $\rho_{1}$, $\rho_{2}$, …, that satisfy the above-mentioned conditions and $0\leq \rho_{k}\leq \frac{1}{2}$, however:*
$$\begin{array}{ccc} \rho_{1} & = & \frac{F_{N-1}}{F_{N+1}}\;,\\ \rho_{2} & = & \frac{F_{N-2}}{F_{N}}\;,\\ & \vdots \\ \rho_{k} & = & \frac{F_{N-k}}{F_{N-k+2}}\;,\\ & \vdots \\ \rho_{N} & = & \frac{F_{0}}{F_{2}}=\frac{1}{2}\;, \end{array}$$
*where $F_{0}=1$, $F_{1}=1$, $F_{2}=2$, $F_{3}=3$, $F_{4}=5$, $F_{5}=8$, $F_{6}=13$, $F_{7}=21$, … are Fibonacci numbers with $F_{k}=F_{k-1}+F_{k-2}$  for the optimal sequence, resulting in maximal range reduction [48,49].*


**Remark** **2.**
*At the N-th iteration, it holds that $\rho_{N}=\frac{1}{2}$, and the two intermediate points are chosen to be the same, making a decision regarding where the optimum lies impossible. To avoid this problem, the N-th reduction factor is modified to $\rho_{N}=\frac{F_{0}}{F_{2}}-\delta =\frac{1}{2}-\delta$, where 0<δ≪1.*


**Proposition** **4.**
*The required number of iterations N to achieve selected accuracy ϵ satisfies $F_{N+1}\geq \frac{1}{\epsilon }$, and the two intermediate points are selected as below [50]:*
(5)$$\begin{array}{ccc} \;\;\;\;\;\;\;{\hat{x}}^{\left(k^{-}\right)} & = & x^{\left(k-1^{-}\right)}+\rho_{k}\left(x^{\left(k-1^{+}\right)}-x^{\left(k-1^{-}\right)}\right)\;, \end{array}$$
(6)$$\begin{array}{ccc} \;\;\;\;\;\;\;{\hat{x}}^{\left(k^{+}\right)} & = & x^{\left(k-1^{-}\right)}+\left(1-\rho_{k}\right)\left(x^{\left(k-1^{+}\right)}-x^{\left(k-1^{-}\right)}\right)\;, \end{array}$$
*where*
(7)$$\begin{array}{c} \rho_{k}=\frac{F_{N-k}}{F_{N-k+2}}\;. \end{array}$$


### 4.4. Sketch of the Algorithm for Optimal Tuning of a Two-Parameter Controller

**Assumption** **2.**
*Suppose we need to tune two parameters of a controller, i.e., $P_{1}$ and $P_{2}$, as an example $k_{P}$ and $k_{D}$ in a continuous-time PD controller.*


**Proposition** **5.**
*The algorithm in the case of a two-parameter controller is composed of the following steps:*
*(0)* 
*calculate/estimate allowable ranges of $P_{1}$ and $P_{2}$ for the given dynamical system, ensuring stability of the closed-loop system (see Remark 3);*
*(1)* 
*define initial value of $P_{2}^{\left(0\right)}$;*
*(2)* 
*using a sequence of Fibonacci-search iterations for defined tolerance ϵ and k=0, implement the following*
***bootstrapping***
*technique (put $P_{2}^{\left(k+1\right)}=P_{2}^{\left(k\right)}$):*
*(2a)* 
*starting with the initial range for $P_{1}$ and fixed $P_{2}^{\left(k+1\right)}$, find by means of the Fibonacci method the optimal ${\hat{P}}_{1}^{*}{}^{\left(k+1\right)}$, and proceed to the step 2b;*
*(2b)* 
*starting with the initial range for $P_{2}$ and fixed $P_{1}^{\left(k+1\right)}={{\hat{P}}_{1}}^{*}{}^{\left(k+1\right)}$, find by means of the Fibonacci method the optimal ${{\hat{P}}_{2}}^{*}{}^{\left(k+1\right)}$, and proceed to the step 2c;*
*(2c)* 
*if the updated point $\left({{\hat{P}}_{1}}^{*}{}^{\left(k+1\right)},\;{{\hat{P}}_{2}}^{*}{}^{\left(k+1\right)}\right)$ has already been found in the past iterations, stop the algorithm (no improvement is possible anymore); otherwise, enter k:=k+1 and proceed to the step 2a.*




The step 2c does not have to be implemented when the number of bootstraps is assumed and the number of iterations *N* of the Fibonacci method is known (see Figure 7). When 2c is omitted, the algorithm has a deterministic running time.

**Remark** **3.**
*Recalling research motivation from Section 3, the proposed method to perform auto-tuning would be even more suitable in the case of fine-tuning of nominal controller parameters aiming at improving its responsive capabilities to different dynamics requirements than to its prototyping from a scratch. The latter is the results of no need to have an analytical criterion using knowledge from a model of UAV, allowing for obtaining a good estimate of the search space for the given controller, ensuring stability of the control system. From the implementation point of view, and in order to ensure the safety of the tuning procedure, it is recommended to start the tuning from narrow deviations with respect to the nominal controller parameters and widen then along the way, repeating the procedure, if necessary.*


**Remark** **4.**
*On obtaining initial ranges for the considered parameters.*

*The step where allowable ranges for the considered parameters is of prime importance and requires the expert knowledge obtained either from a series of flights with a set of different gains, and conclusions drawn from them, or any analytical model of the UAV available. At this point, however, the method would require any initial knowledge concerning the model of the used UAV unit. When fine-tuning is considered, this model is usually not needed, which has been verified by our experiments.*

*In order to obtain a range for the considered parameters, the following approaches might be adopted:*
*(1)* 
*obtain a nominal model of the UAV, e.g., for altitude changes only, as one of [51], include uncertainty information, see [52,53,54,55], and perform stability analysis of the uncertain system, yielding estimates of safe ranges, stabilising the closed-loop system,*
*(2)* 
*for a simulation-developed model of the UAV, as used in the paper in ROS-Gazebo environment, obtained, e.g., by designing a model of the UAV in such software as Autodesk Inventor Professional 2015 (Dyson, 9.0.23.0, Autodesk, San Rafael, CA, USA), perform a series of simulations to observe the behaviour of the model in a simulation environment, as in [56],*
*(3)* 
*obtain a linearized model of the UAV for the expected motion, assume possible disturbances, such as wind gusts, as bounded, an obtain stability analysis as in [57] considering ranges for controller parameters,*
*(4)* 
*heuristic approach: set the ranges close to the nominal parameters of the controller, which proved to stabilise the UAV in real-world experiments.*


*The stability analysis is a major requirement to fly the UAV safely in any environment, and should be performed prior to performing any experiments. As stated, however, above, it does require any knowledge about the model, in such an approach.*

*On the other hand, from the adaptive control viewpoint, based on the works of [58,59], one can use dynamic programming techniques, presented also in [60], to perform optimality and stability analyses.*


### 4.5. Stopping Criteria for Iterative Algorithms

**Proposition** **6.**
*As a part of the step 2c from the previous algorithm, the following sample stopping criteria might be potentially considered:*

*theoretical tests (in general, for $\underline{x}\in R^{n}$)*
(8)$$|f\left({\underline{x}}^{\left(k\right)}\right)-f\left({\underline{x}}^{*}\right)|<\epsilon_{1},\;∥{\underline{x}}^{\left(k\right)}-{\underline{x}}^{*}∥<\epsilon_{2}\;,$$

*practical stopping criteria*
(9)$$|x_{i}^{\left(k+1\right)}-x_{i}^{\left(k\right)}|\leq \epsilon_{i}$$
*or*
(10)$$|f\left({\underline{x}}^{\left(k+1\right)}\right)-f\left({\underline{x}}^{\left(k\right)}\right)|<\epsilon_{1}\;,$$

*where ${\underline{x}}^{\left(k\right)}$ denotes the solution at k-th iteration of the algorithm, and ${\underline{x}}^{*}$ is the true minimum of the cost function $f(\underline{x})$.*


However, in this approach, the number of iterations is limited by the expected number of bootstraps and expected accuracy of the result, which are functions of the duration of the experiment.

### 4.6. Outline of the Implementation of the Algorithm

The method is iterative-based and collects information about the performance index (on incremental cost function value) at sample time instants, equally spaced every $T_{S}$ seconds. Assuming the sampling period is sufficiently small, a single main iteration of the method is initialized $N_{\max}$ times where:for $i=1,\;\ldots ,\;N_{c}-1$ and with the controller parameters updated in the previous iteration, the performance index is evaluated as
(11)$$J^{\left(i\right)}=J^{\left(i-1\right)}+\Delta J^{\left(i\right)}\;,$$
where $\Delta J^{\left(i\right)}$ might be, e.g., chosen as $\Delta J^{\left(i\right)}=\left|e_{i}\right|$ corresponding to the *i*-th sample of the tracking error at time $t=iT_{S}$ or $\Delta J^{\left(i\right)}=e_{i}^{2}+u_{i}^{2}$ corresponding to the sum squared values of the tracking error and control input to the plant;for $i=N_{c}$, a single iteration of the Fibonacci method is initialized—the cost function (performance index) value is stored, and if it is possible to compare the values of cost functions in a given range, i.e., either if the two intermediate points have been evaluated, the range has been reduced or the optimal solution has been found and bootstrapping takes place—either way at this point controller parameters change (are updated), which results in transient behavior of the dynamical system,for $i=N_{c}+1,\;\ldots ,\;N_{\max},$ no action is taken (the parameters of the controller have been updated, no performance index is collected), and these steps are intended to allow the closed-loop system to stabilize at some point (to decay the transients), in order to allow performance index evaluation during the next main iteration (starting again from i=1).

Thus, the experiment-based tuning process takes $N_{\max}\cdot T_{S}$ seconds, which should be correlated with the reference signal changes. Having assumed that a complete run for the Fibonacci method takes *N* steps and two parameters are changes, every bootstrap stage takes $2N\cdot N_{\max}\cdot T_{S}$ seconds. This time horizon can be reduced if past the first iteration of the Fibonacci method, a single intermediate point is evaluated only, with the other taken from one of the previous iterations.

## 5. Experimental Platform

The experiments have been carried out using a customized micro aerial vehicle platform (see Figure 1). The considered UAV composed of a DJI hexacopter frame F550 (Shenzhen, China) and E310 DJI motors, UAV PixHawk Autopilot, Mobius ActionCam Lens C2 and Matrix Vision mvBlueFOX-MLC200w cameras, Real Time Kinematic GPS, Rangefinder TeraRanger One sensor and an onboard Intel NUC-i7 PC (Santa Clara, CA, USA) has been developed for the purposes of the Mohamed Bin Zayed International Robotic Challenge 2017, where its high performance has been proven in Treasure Hunt and Landing tasks [8,61].

In the present experiments, the *Gripper* module is detached. Its use in carrying load will be a subject of further, separate, tests. It is also planned in the future to apply the FGT method to automatically tune the parameters of the altitude controller and the parameters of orientation controllers (roll, pitch, and yaw angles) during flight with attached load with different masses. However, this work focuses on automatic and minimum-seeking tuning of parameters of an altitude controller in a UAV during flight to track reference signals with different dynamics, named, respectively, slow (S), medium (M), fast (F)—see Figure 8. As a result, and having applied the FGT method, three sets of controller parameters are obtained which can be used to translate between them to obtain smooth and stable altitude changes. This is particularly useful when a sharp picture has to be obtained from video cameras to, subsequently, analyze it or to perform fast autonomous landings from a set altitude, e.g., on a moving vehicle, as in Figure 1b.

By performing the fusion from on-board sensors, an estimation of the UAV states can be performed, which is an important part in this UAV system to enable its precise and reliable control at high speeds. The current architecture is based on a combination of an Extended Kalman Filter (EKF) which is implemented in PixHawk, gathering the data from GPS and the Inertial Measurement Unit (IMU), magnetometer, and barometer, as well as the output from the Rangefinder using a Linear Kalman Filter (LKF) in the onboard computer.

Since the control system must be capable of ensuring good performance at large deviations from the hover configuration or to compensate heavy wind, a nonlinear controller is used. The UAV has torque $\underline{m}\in R^{3}$ and thrust force control f∈R as inputs, thus based on [62,63]
(12)$$\begin{array}{ccc} \underline{m} & = & -k_{R}{\underline{e}}_{\;R}-k_{\Omega }{\underline{e}}_{\;\Omega }+\underline{\Omega }\times J\underline{\Omega }-J\left(\hat{\underline{\Omega }}R^{T}R_{c}{\underline{\Omega }}_{\;c}-R^{T}R_{c}{\dot{\underline{\Omega }}}_{\;c}\right)\;, \end{array}$$
(13)$$\begin{array}{ccc} f & = & -\left(-k_{P}{\underline{e}}_{\;x}-k_{ib}R\int_{0}^{t}R{\left(\tau \right)}^{T}{\underline{e}}_{\;xd}\tau -k_{iw}\int_{0}^{t}{\underline{e}}_{\;xd}\tau -k_{D}{\underline{e}}_{\;v}-mg{\underline{e}}_{\;3}+m{\ddot{\underline{x}}}_{\;d}\right)R{\underline{e}}_{\;3}, \end{array}$$
where $J\in R^{3\times 3}$ is the inertia matrix with respect to the body frame, $R\in R^{3\times 3}$ is the orientation of the center of mass matrix, $R_{c}\in R^{3\times 3}$ is the commanded value of the inertia matrix with respect to the body frame, $\underline{\Omega }\in R^{3}$ is the angular velocity vector in the body fixed frame, ${\underline{\Omega }}_{\;c}\in R^{3}$ is the commanded angular velocity vector in the body fixed frame, ${\ddot{\underline{x}}}_{\;d}$ the desired acceleration, and $k_{P}$, $k_{D}$, $k_{R}$, $k_{\Omega }$ positive definite terms.

The $e_{x}$, $e_{v}$, $e_{R}$, and $e_{\Omega }$ denote, respectively, position, velocity, orientation, and angular rate errors, with the subscript $\cdot_{C}$ denoting the commanded value. In our approach, the gains $k_{P}$ and $k_{D}$ are subject to tuning.

The control design, i.e., the controller, is a well-known structure, compensating multiple nonlinear effects known from UAV dynamics, and previously described in multiple papers of the authors, e.g., [8,61]. It has been assumed that, in order to stress the main point of the paper, and its main novelty, the section presenting the controller has been reduced to the necessary minimum. The controller design does not result from control point of view, but from the nonlinear dynamics of the plant.

The combination of ROS and the Gazebo environments has also been used at the stage of prototyping the solutions and at the stage of simulation tests of the FGT methods. Gazebo under ROS is a respected robotic simulator, which can be used for simulation-in-the-loop (SITL) together with a firmware from PixHawk. The results of simulation and experimental tests selected and presented in the paper, to the best knowledge of the authors, are most representative, and are described later in the text. The experiment with a real robot and a computational time analysis reflects the applicability of the FGT method to real problems. A video reporting one of the experiments is attached to this paper and can be found in the Mathematical Models Database (http://mathematicalmodels.put.poznan.pl) [64]. In addition, it is planned in the near future to make the overall system with implementation of all method variants public in the MMD together with numerous data sets available as ROS-files for testing the system in a Gazebo simulator. Further information regarding hardware used is available in the previous author’s papers, i.e., [8,65] and in previously self-cited papers, therein. A detailed description of hardware and software solutions is presented there, including path planning, low- and high-level control, image recognition, etc. In addition, one may find therein an extended description of the implemented and tested precise multi-level estimator of the altitude of the UAV, which fuses data from GPS and a laser Rangefinder sensor to enable flying smoothly above a vehicle and landing on it. Finally, the authors provided a description of how they integrated an estimator of velocity and position of the vehicle into the MPC controller of the UAV to be able to achieve fast landing on a moving object. The developed approaches and gained experience during the MBZIRC 2017 provide the starting point for gaining results from the tests in further parts of the text.

## 6. Experimental Results

### 6.1. Introduction

In this section, the performance of the FGT method as well as its experimental verification are presented (for various scenarios), referring to the theoretical results from Section 4.

### 6.2. ROS Implementation of the FGT Algorithm

For the model of the DJI F550 hexacopter developed in Gazebo, the described FGT algorithm has been implemented under the ROS control for tuning of the two parameters, namely $k_{P}$ and $k_{D}$, given their ranges and the initial value of the second parameter. These values have been taken on the basis of examination of the results of manual tuning of the UAV in manual flight conditions.

In real world applications, in a disturbed environment, and when control signals are constrained, as well as the feasible dynamics of the UAV, the most appealing control law from the theoretical point of view might give unacceptable results when the gusts of wind will be greater than those claimed to be in safe ranges. The authors do not perform here any stability analysis, making an assumption, that the ranges for sought parameters are ’safe’, in the sense that any combination of them does not cause the closed-loop system to become unstable. This is a realistic assumption, as operators of UAVs usually try to fly their machines selecting a set of controller parameters, and based on them, a smallest set of gains can be formulated, leading to searching the locally-best gains between the stabilizing parameters of the controller.

The main loop of the program is executed with the frequency of 100Hz, whereas the tuning algorithm with 5Hz (i.e., every 200ms). The syntax of the method is as follows:
update_parameters_FIB(input,main_iteration_counter,P1_range,P2_range,N_c,N_max,Par_initial,method,output)
with the following parameters:input—the current value of the increment of the performance index, i.e., the tracking error or the low-pass filtered tracking error;main_iteration_counter—the variable referring to the number of iterations of the algorithm, here: 48;P1_range—the range for the first tuned parameter, here [4,12];P2_range—the range for the second tuned parameter, here [2,7];N_c—the number of samples when the performance index is collected (see Figure 9), here 50;N_max—the number of samples corresponding to the length of the trajectory primitive, here 60;Par_initial—the initial value for the second parameter, here treated as variable;method:0—the performance index is calculated at every step;1—the performance index is averaged over all past measurements with the same parameters;2—the performance index is evaluated only for parameters that have been unconsidered before (the length of the tuning procedure is reduced);output—the vector of output variables, including tuned gains.

The designed method has the following features:comprises two bootstraps;ϵ=0.05 (see Figure 10);the number of iterations in a single bootstrap is 2×2×6=24 (each iteration is composed of two evaluations of performance indices);δ=0.01 (Fibonacci–search);requires 48 loops of a single trajectory primitive.

In order to change the trajectory primitive, the N_max multiplicity of the time interval (multiplicity of the inverse of the frequency of executions of the tuning method) must be equal to the duration of the trajectory primitive. In this way, it can be changed, e.g., extended, for other regimes of work of the UAV, such as tuning at high speeds.

For example, when the trajectory primitive has 12s duration and the experiment consists of 48 iterations (two bootstraps in two parameters), the total length of the experiment is 576s. For the sampling period $T_{S}=0.2\;s$, there are 60 samples for one period of the trajectory primitive ($N_{c}=50$, $N_{\max}=60$). The sample experimental result can be found in the Appendix B. In Appendix C, a detailed explanation of the tuning process from the first iterations is included.

### 6.3. ROS Gazebo Simulation Results

To mimic the test presented in Section 6.2 (see Figure 11), a few hundred simulation results have been gained in various configurations of the FGT algorithm, with results of 90 tests selected (presented in Table 2). It must be noted that some initial values of the parameters have been intentionally selected as out of range to show the proper and desired behavior of the FGT method. These results form a base to formulate comments concerning the procedure and draw conclusion, and serve as a good starting point for conducting experiments on a real UAV. The FGT algorithm has been tested in different configurations and variants with great attention paid to the efficiency of obtaining PD controller parameters and the performance of the obtained tracking properties with the reference altitude supplied to the control system. The following have been tested:(i)accuracy of the obtained controller parameters for reference primitives with different dynamics: fast ($v_{\max}=10\;\frac{m}{s}$, $a_{\max}=100\;\frac{m}{s^{2}}$), medium ($v_{\max}=3.33\;\frac{m}{s}$, $a_{\max}=11.11\;\frac{m}{s^{2}}$), slow ($v_{\max}=1.67\;\frac{m}{s}$, $a_{\max}=2.78\;\frac{m}{s^{2}}$) reference primitives,(ii)the impact of the initial value of the second parameter of the controller (Par_initial) on the convergence of the FGT procedure,(iii)ability to reduce tuning errors using a low-pass first-order filter in the variants listed above where a low-pass filter recursive equation is proposed to diminish the impact of the measurement noise on the results of optimization.

Each version of the FGT procedure has been tested in two ways: with and without low-pass filtration, which gives six independent results (versions) of the FGT method. The low-pass filter is assumed to have the transfer function
(14)$$\begin{array}{c} G\left(s\right)=\frac{1}{1+0.1s}\;, \end{array}$$
and to be discretized with sampling period $T_{S}=0.01\;s$. The resulting recursive equation of the low-pass filter is given as
(15)$$\begin{array}{c} y\left(n\right)=a\left(n-1\right)+\left(1-a\right)u\left(n-1\right)\;, \end{array}$$
where the constant parameter
(16)$$\begin{array}{c} a=\exp\left(-T_{S}/0.1\right)\;, \end{array}$$
with y(n) and u(n) as filtered and pure errors at sample *n*, respectively.

In all of the 90 presented results (remaining configuration parameters as in Section 6.2), the incremental cost function has been the sum of absolute values of the tracking error:(17)$$J=\sum_{i=1}^{N_{c}}\left(\right|e_{i}\left|\right)\;.$$

Summing up cost function values from Table 2, the lowest *J* is equal to 9.227 and has been obtained in the configuration 1–F–1 (method–reference primitive–filtration) for all three reference primitives and different initial points for the second parameter. This configuration of the method has superior performance in comparison with the others. The second best is the 2–F–1 configuration, when both dynamics M and F have given better tuning results than S.

The configurations listed in the above paragraph perform much worse in the case of no low-pass filtration (the highest total cost function values are 12.041 and 11.261, respectively). By performing low-pass filtration, the results (*J*) improve by 16.80%, giving 20 superior results in the case of filtration in comparison with two superior results in the case of no filtration. Similarly, seven worst results in comparison with 16 worse results in the case of no filtration. It is to be noted that omitting the filter deteriorates the results for F dynamics, and, in addition, the algorithm is more sensitive to the case when the initial point is out-of-defined-range.

On the basis of histograms of the final values of the following tuned parameters $k_{P}$ and $k_{D}$ presented in Figure 12, Figure 13 and Figure 14, it can be said that the controller parameters for the defined reference trajectory primitives result in similar values of tuned gains, where small differences between the parameters inform us that either the true minimum of the cost function is relatively flat, and extend some of the parameter ranges, or that the function has multiple minima. Differences between cost functions *J* from the last iterations result from disturbances acting on the system.

Using a low-pass filter for the measured tracking error signal, the method is less vulnerable to measurement noise, and the results might be treated as more accurate.

Overall results are close to those obtained in the configuration with a low-pass filter. Using unfiltered measurement data results in flat histograms, and instead of a single pair of controller parameters repeatedly obtained among simulations, neighboring parameters tend to be seen for every experiment. A normal-like density function (Gauss-like) can be observed in the case of values for $k_{D}$, which could also be true for $k_{P}$; however, the upper value of the admissible range for this parameter turned out to be too small.

In further tests, the following configuration of parameters has been chosen:method = 1,filter = 1,Par_initial = 1,
with the remaining conditions as presented in Section 6.2 and in Figure 11. Additional 30 tests have been carried out where every case has been tested 10 times for all three trajectory primitives (S, M, F). Table 3 presents the results of analysis of these tests based on the obtained data from ROS-Gazebo. Figure 15 depicts the results obtained from all three series of experiments, i.e., plots of functions $J=f(k_{P},\;k_{D})$ and their surface approximations.

### 6.4. Tests in Flight Conditions

For the purpose of this publication, sample results have been obtained during one of the verification experiments carried out to autotune altitude controller parameters using the FGT method of the DJI F550 hexacopter in flight conditions. Real outdoor scenario experiments, analogous to these presented in Section 6.3 (see Figure 11), have been carried out using a real machine this time during an intensive experimental camp in the countryside of South Bohemia (the Czech Republic) (see the website: http://mrs.felk.cvut.cz/projects/mbzirc which is systematically updated by its authors with videos and photos documenting the experiments carried out during these camps). In the outlined experiment, the same configuration of the FGT as in Section 6.3 has been used, where in the following case:method = 1,filter = 1,Par_initial = 1,trajectory primitive (M).

The total time of a single experiment is 11 min, with 9.6 min taken by two full bootstraps of the Fibonacci-search algorithm. From all the results presented in the paper, it can be seen that the choice of the number of bootstraps, the number of main iterations and the choice of the accuracy of the Fibonacci-search method are not random and can be determined from the constraints resulting from the choice of voltage source. Based on the electrical capacity of the batteries used in the selected hexacopter UAVs, the maximum time of flight has been estimated for the planned tests. This allowed to perform two full bootstrap sequences. It has been assumed that the voltage level drop during the experiments is negligible, if the batteries are initially fully charged. It is a natural simplification leading to the assumption that the thrust force of the driving units, dependent on the voltage level, is constant during the flight. Due to the fact that in the selected optimization method, the time of termination of the optimization procedure is deterministic, the autotuning experiment can be divided into stages (bootstraps), which can be carried out independently. This allowed for minimizing the impact of the voltage drop on the tuning results.

In the discussed trial, the main point has been the evaluation of the convergence of the algorithm during the experiment, expressed by a reduction of the cost function after every iteration of the Fibonacci-search algorithm after the tuned parameter has been amended. To depict the performance of the FGT algorithms, the complete data set is presented in Figure 16, Figure 17 and Figure 18 obtained from the considered trial. The first bootstrap is particularly interesting (see Figure 16 and the output presented in the Appendix) where one can see that starting from iteration no. 10, the cost function value reduces to the obtained minimum every few iterations. In addition, it is obtained with totally different PD controller parameters, which supports previous statements concerning the properties of the cost function $J=f(k_{P},\;k_{D})$ (Figure 15).

A set of pictures taken during the real-world experiment, in windy conditions, is presented in Figure 19. A sample clip from this tuning can be found in the Mathematical Models Database (http://mathematicalmodels.put.poznan.pl) [64].

## 7. Conclusions

In the paper, a new real-time model-free minimum-seeking autotuning method for controllers based on the Fibonacci-search algorithm is presented. This gain tuning strategy is introduced in the context of improving reference signal tracking during flight in multirotor UAVs. As discussed in the introduction, due to the variety of tasks or missions of UAVs, one can distinguish multiple dynamic conditions of flight. In these conditions, controller parameters must ensure superior performance of the control system. Simple, fixed-value controllers can be successfully used, provided that they are well-tuned. In this context, the FGT method should be of the greatest use. The following may be included among the most important properties of this algorithm:it allows automatic tuning, in predefined ranges, of parameters of a controller that is widely used with a low number of parameters, such as of PD or PID type (if the method is extended to 3-parameter framework, which is not discussed in the paper, though, is the topic of the current research);the resulting solution from the FGT algorithm is optimal in the sense of the optimality of a function, with respect to the prescribed accuracy, and the cost function might be, in general, arbitrarily chosen;with respect to multirotor UAV altitude controllers, the optimal tuning process is carried out in real time and can be conducted either using a model of a UAV, or using a real robot—to the best knowledge of the authors, there are no such results reported in the literature up until now;two-parameter controller tuning due to the use of the proposed Fibonacci-search algorithm is characterized with rapid convergence, as has been observed during experiments even past several iterations of the algorithm; the obtained controller parameters ensure visible improvement in the reference altitude tracking task in comparison to nominal controller parameters; the time required to obtain optimal parameters is considerably shorter in comparison to bio-inspired batch methods;due to the structure of the algorithm, the FGT method can be used in a single run or performed in stages, in separate flights taking the maximum flight time limited by capabilities of the voltage source in a particular UAV; from authors’ observations, it can be said that provided the UAV has a maximum time of flight approximately equal to 20 min, and the tuning experiment takes about 9.5 min to terminate (576s) for smooth reference altitude tracking, the experiment can be performed in a single run; however, if a visible voltage drop is observed, resulting in a loss in the effective thrust force of the used driving units, the FGT approach should be used in stages, e.g., the first bootstrap in the first stage, and the second one in the next flight with new batteries;the last and obvious feature of the FGT algorithm that has appeared as the result of the experiments is the safety of this method during autotuning process of the altitude controller of the hexacopter UAV in flight conditions; in all the tests, independently of the initial value of the second parameter, no stability loss of the UAV has been noted, which could be a potential threat for the operator, environment or the very robot; naturally, any initial/expert knowledge concerning nominal parameters of the controller that can be used during flight is of importance. These parameters can be fine-tuned in selected ranges using the FGT method; intuitively, to obtain these ranges for parameters can be extended in consecutive experiments or to use any rapid prototyping methods prior to the experimental stage.

This work has successfully demonstrated the capabilities, simplicity, utility and applicability of the FGT method. The above list presents the main contributions of this paper from the perspectives of UAV control. Furthermore, the conducted experiments show the performance of the FGT method and verify its basic properties in simulated and real environment. Challenges identified during the flight-tests have been outlined and discussed. Considering the six proposed variations of the method, the superior case is with the method equal to 1, using a low-pass filtration and where the performance index is averaged over all past measurements with the same parameters. This case is widely-discussed in the paper and presented in a statistical analysis based on the results obtained from ROS-Gazebo simulations, and with representative results obtained in real experiments. The latter results have been obtained in real outdoor scenarios during flight of a real hexacopter UAV. This variant of the FGT method has been equipped with a low-pass filter recursive equation to diminish the impact of the measurement noise on the results of optimization. During the experiments and simulation tests, the superior performance has been obtained for this particular configuration which highlighted a number of directions for future research. In particular, it is planned to compare this approach to the FGT method with other optimization methods based on golden-search and bisection algorithms.

This work was motivated by the MBZIRC 2017 competition and mainly by the challenges requiring precise altitude control, such as autonomous landing and object grasping in demanding outdoor conditions. In addition to the competition tasks, the results of this research are important in further exploitation of the MBZIRC system in multi-UAV tasks and in applications where a manual tuning approach cannot be used (for example, see [66]).

## Figures and Tables

**Figure 1 sensors-19-00312-f001:**
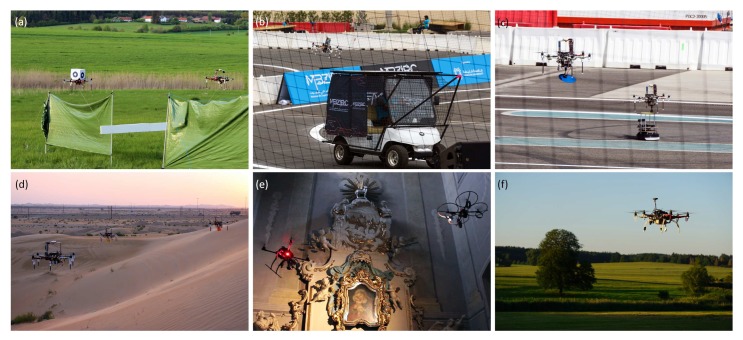
Six-rotor UAV used during experiments: (**a**) cooperative transport of a large object motivated by the third challenge of the MBZIRC 2017 competition (https://youtu.be/rTlZQ1x93h0); (**b**) autonomous landing on a driving vehicle (https://youtu.be/zHs5LtLyBsc); (**c**) robots cooperating in a transport task (https://youtu.be/ogmQSjkqqp0?t=205); (**d**) surveillance mission of a UAV formation; (**e**) recording a video inside a historical building by a pair of UAV (a cameraman robot, and a lightning robot) (https://youtu.be/-sTUwzFf_Mk); (**f**) outdoor experiment in the countryside of South Bohemia (Czech Republic) during one of the testing days for the MBZIRC 2017 competition.

**Figure 2 sensors-19-00312-f002:**
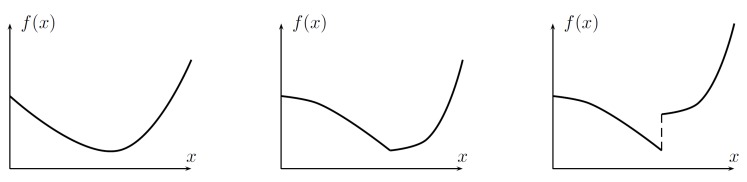
Examples of unimodal functions.

**Figure 3 sensors-19-00312-f003:**
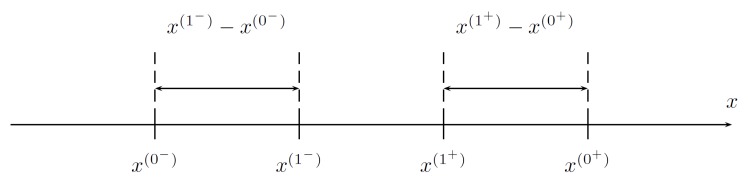
Two intermediate points in the considered range.

**Figure 4 sensors-19-00312-f004:**
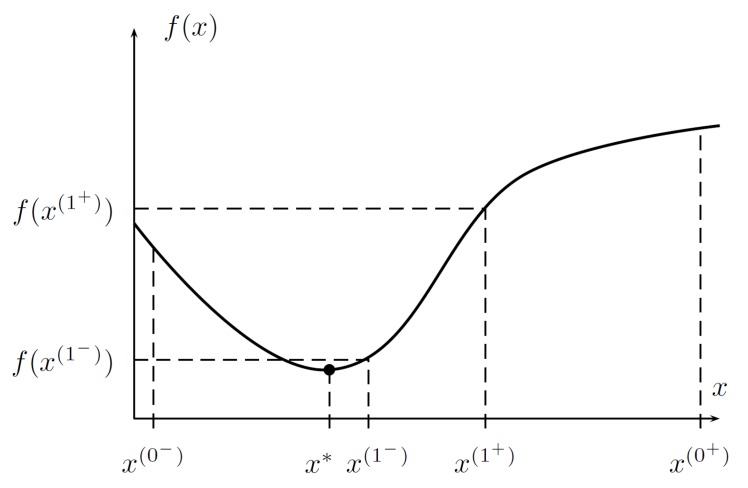
The case for $f\left(x^{\left(1^{-}\right)}\right)<f\left(x^{\left(1^{+}\right)}\right)$, where the minimum $x^{*}\in \left[x^{\left(0^{-}\right)},\;x^{\left(1^{+}\right)}\right]$.

**Figure 5 sensors-19-00312-f005:**
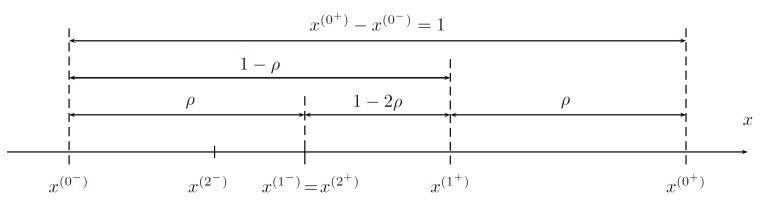
Obtaining ρ to enable evaluation of *f* at a single point only.

**Figure 6 sensors-19-00312-f006:**
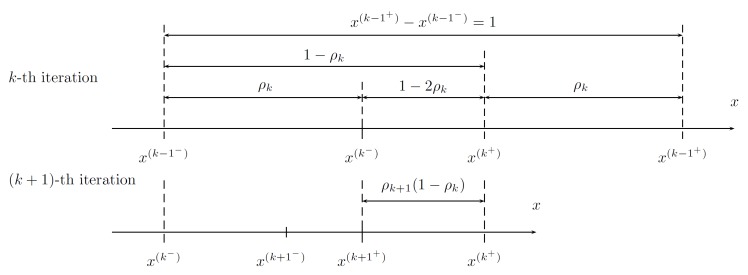
Selection method of the intermediate points, ensuring property discussed in Proposition 3.

**Figure 7 sensors-19-00312-f007:**
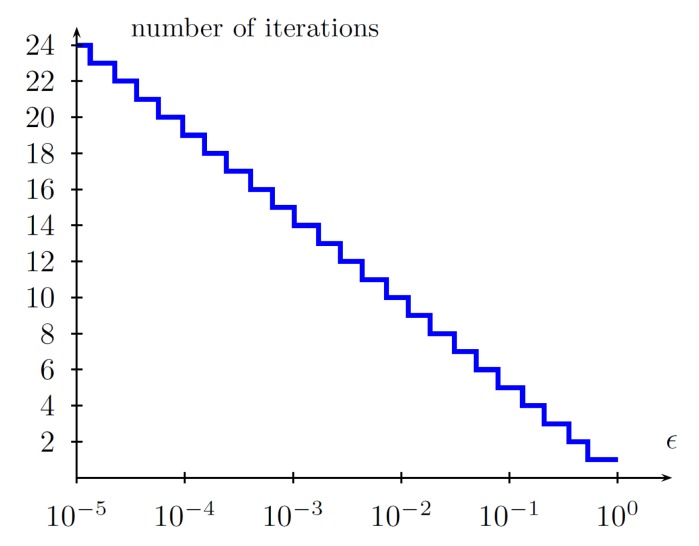
Number of iterations in a Fibonacci-search algorithm vs. ϵ.

**Figure 8 sensors-19-00312-f008:**
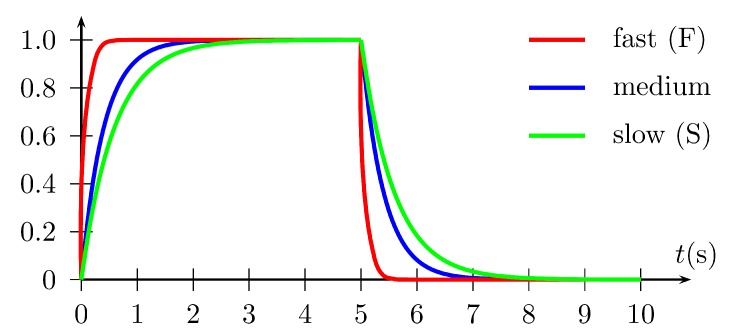
Trajectory primitives used during experiments (see the text for further information).

**Figure 9 sensors-19-00312-f009:**
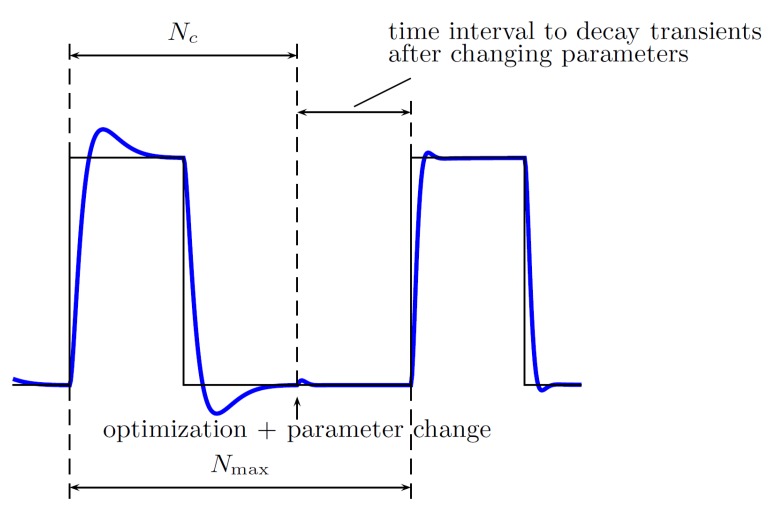
Values of $N_{c}$ and $N_{\max}$ must be chosen in the way allowing transients to decay, after controller parameters are changed by the FGT method (depicted: reference vs. actual output, evolution in time).

**Figure 10 sensors-19-00312-f010:**
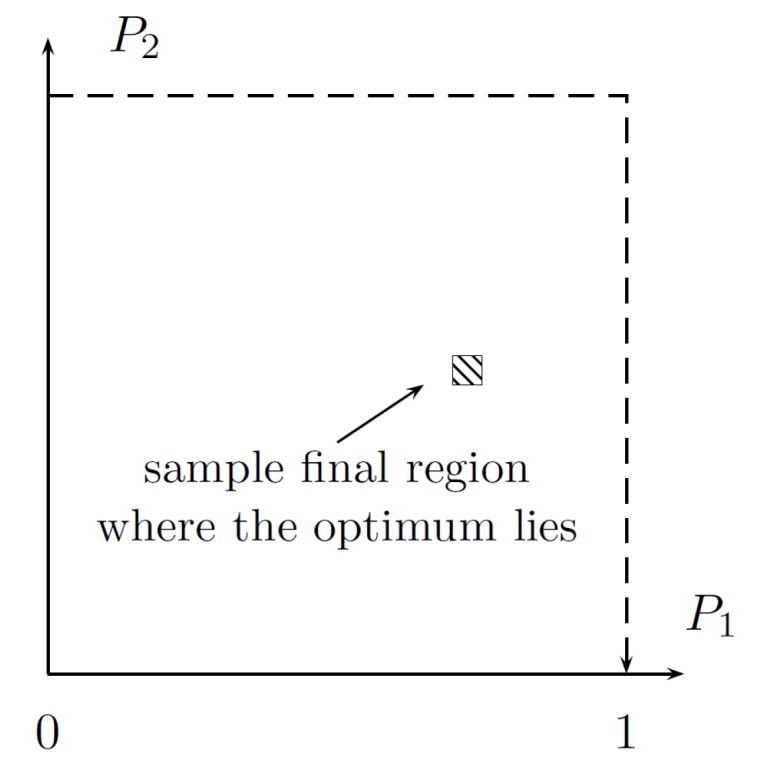
Visualization of ϵ=0.05 accuracy for sample arbitrary ranges [0,1] of the both parameters.

**Figure 11 sensors-19-00312-f011:**
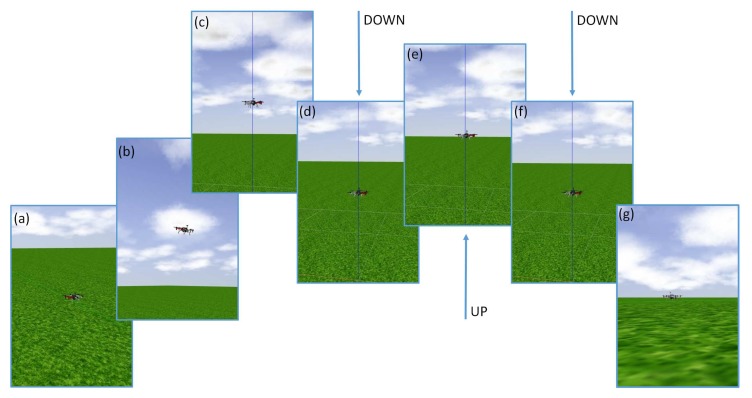
Sample snapshots of one of the experiments conducted in ROS: (**a**,**b**) UAV takes off from the ground and flies to the desired position in an autonomous mode; (**c**) safe altitude is reached (in desired coordinates); (**d**) decreasing the height down to 2m; (**e**,**f**) altitude controller (PD) autotuning: repeated 48 cycles of raising up to 3m and going down again the level of 2m (12s per single iteration); (**g**) termination of the autotuning experiment with autonomous landing of the UAV on the ground (http://mathematicalmodels.put.poznan.pl).

**Figure 12 sensors-19-00312-f012:**
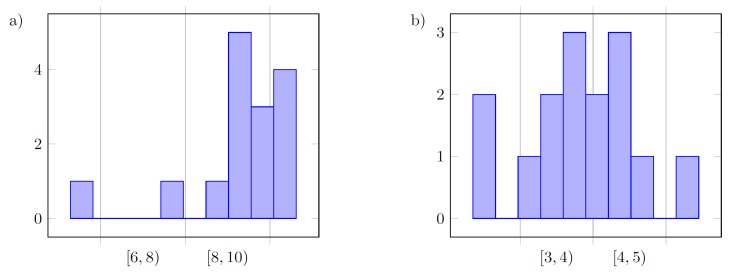
Histogram of the final values for the following parameters in the case without filter (F): (**a**) $k_{P}$; (**b**) $k_{D}$, for all three methods (15 values).

**Figure 13 sensors-19-00312-f013:**
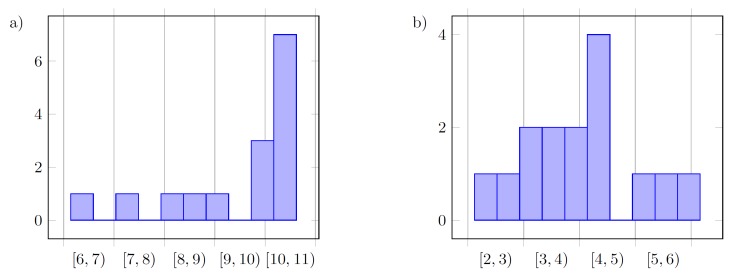
Histogram of the final values for the following parameters in the case with a low-pass filter (F): (**a**) $k_{P}$; (**b**) $k_{D}$, for all three methods (15 values).

**Figure 14 sensors-19-00312-f014:**
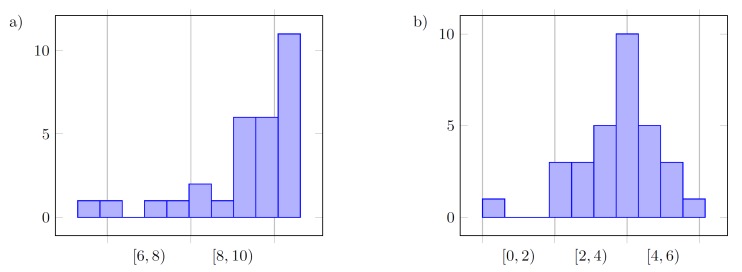
Overall histogram of the final values for the following parameters in the case without filter (F): (**a**) $k_{P}$; (**b**) $k_{D}$, for all three methods and both configurations of the filter (30 values).

**Figure 15 sensors-19-00312-f015:**
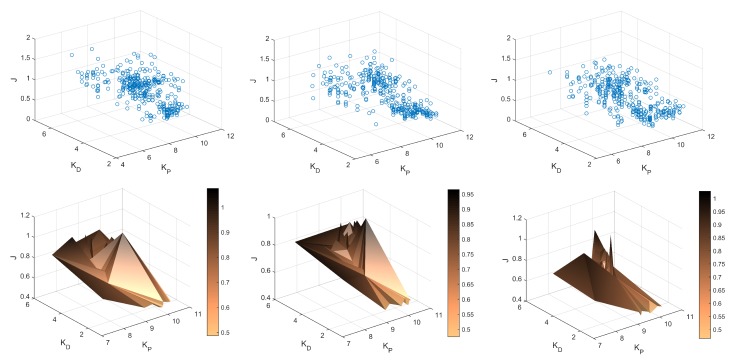
Results obtained from 30 additional experiments carried out in ROS-Gazebo: the plots of functions $J=f(k_{P},\;k_{D})$ and their surface approximations for fast (left column), medium (central) and slow (right) trajectory primitives.

**Figure 16 sensors-19-00312-f016:**
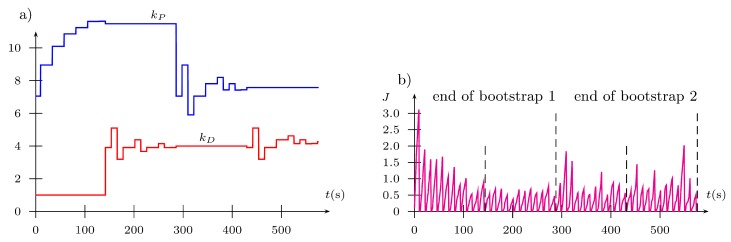
Tuning of controller parameters: (**a**) gains; (**b**) performance index.

**Figure 17 sensors-19-00312-f017:**
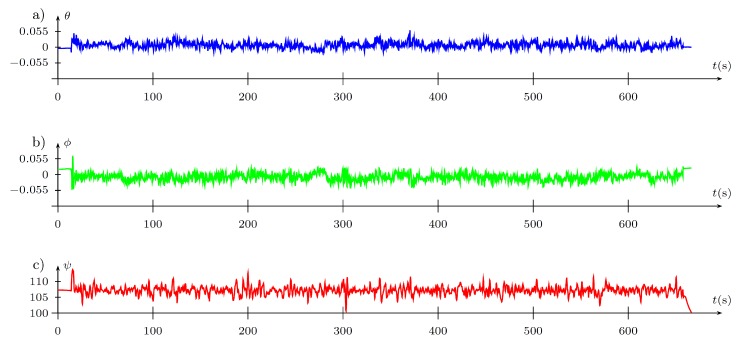
(**a**) pitch θ; (**b**) roll ϕ; (**c**) yaw ψ angles in (${}^{\circ }$).

**Figure 18 sensors-19-00312-f018:**
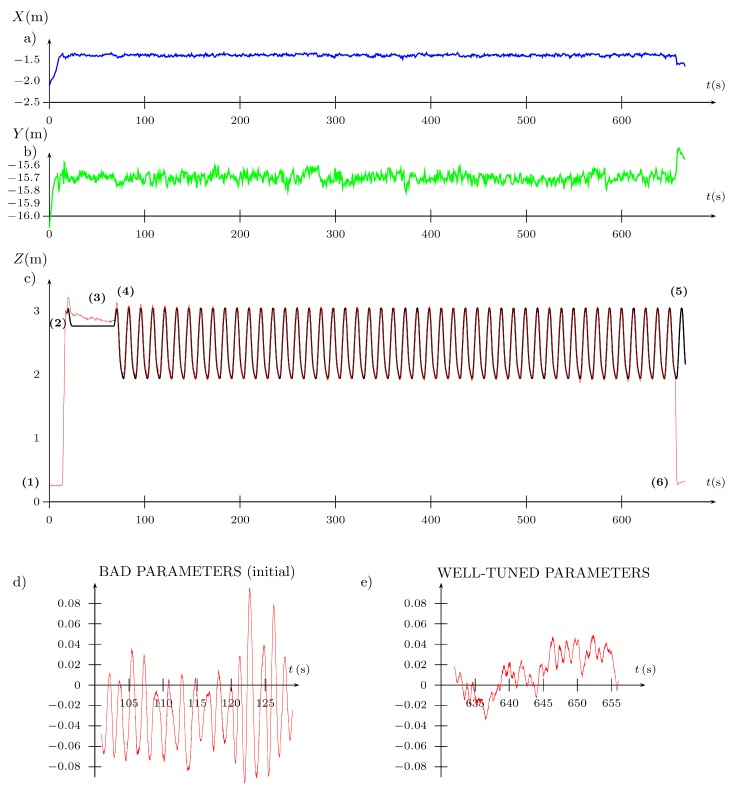
Position in the world frame (m), in the axis: (**a**) *X* (East); (**b**) *Y* (North); (**c**) *Z* (UP) relative to a fixed point in space. Phases of flight: (1) arming the UAV; (2) reaching the desired position; (3) attaining the desired altitude; (4) tuning starts; (5) tuning stops; (6) landing; (**d**,**e**)—error between commanded and actual *Z* (UP) positions.

**Figure 19 sensors-19-00312-f019:**
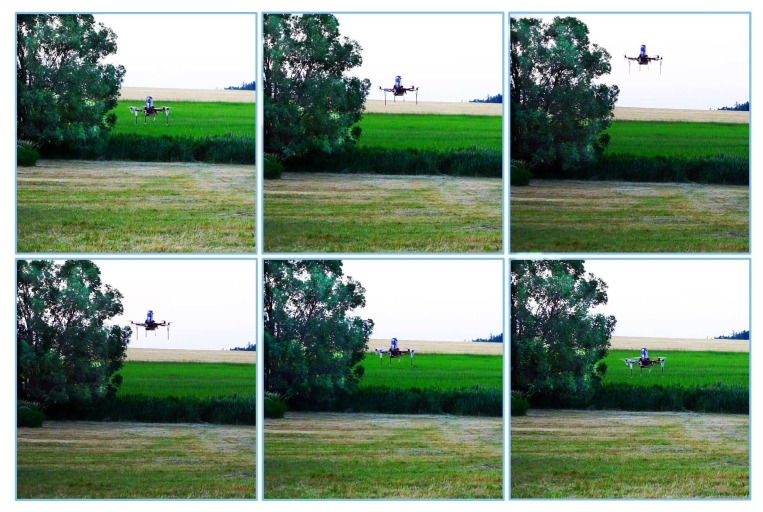
Sample photos from the outdoor experiment in windy conditions.

**Table 1 sensors-19-00312-t001:** Notation in the paper.

$x^{\left(k^{-}\right)}$	lower bound for a parameter at *k*-th iteration
$x^{\left(k^{+}\right)}$	upper bound for a parameter at *k*-th iteration
$\hat{x}$	candidate point in branch-and-bound procedure
f(·)	cost function
$D^{\left(k\right)}$	considered range of a parameter at *k*-th iteration
${\hat{x}}^{*}$	iterative estimate of the optimal solution

**Table 2 sensors-19-00312-t002:** Tuning results from 90 experiments carried out in ROS-Gazebo.

			Slow (S)			Medium (M)			Fast (F)		
method	filter	Par_init	$k_{P}$	$k_{D}$	*J*	$k_{P}$	$k_{D}$	*J*	$k_{P}$	$k_{D}$	*J*
0	0	1	8.34	5.45	1.07	11.76	2.36	0.75	9.86	3.55	0.61
0	0	3	8.34	4.50	1.01	9.48	2.36	0.67	10.62	4.03	0.88
0	0	5	4.50	9.86	0.85	10.62	4.03	0.87	10.62	5.45	0.78
0	0	7	10.62	6.16	0.64	11.00	4.50	0.73	9.86	2.35	0.68
0	0	9	9.96	5.69	0.73	9.95	4.26	0.70	9.86	3.55	0.73
**Total *J* for S, M, F:** 11.713											
0	1	1	7.57	3.55	0.80	8.04	3.21	0.53	9.86	6.16	0.68
0	1	3	10.62	2.12	0.52	5.76	2.84	1.15	9.86	4.50	0.75
0	1	5	6.43	4.26	0.78	10.62	2.12	0.46	10.33	4.02	0.56
0	1	7	9.48	4.74	0.73	9.95	3.78	0.56	10.62	3.55	0.60
0	1	9	11.76	5.93	0.66	10.33	4.02	0.72	9.86	5.45	0.85
**Total *J* for S, M, F:** 10.368											
1	0	1	8.72	4.74	0.96	8.72	2.59	0.55	10.62	2.36	0.59
1	0	3	10.24	3.07	0.76	10.33	4.26	0.88	5.29	4.26	0.83
1	0	5	6.05	2,59	0.91	10.24	2.35	0.63	10.62	3.79	0.79
1	0	7	9.10	2.83	0.60	8.81	5.93	1.16	9.10	3.78	0.80
1	0	9	8.72	2.60	0.94	9.48	2.60	0.76	9.48	3.79	0.86
**Total *J* for S, M, F:** 12.041											
1	1	1	7.28	4.50	0.94	10.33	4.02	0.61	10.62	4.26	0.63
1	1	3	10.62	2.12	0.43	10.24	3.31	0.46	9.10	3.31	0.66
1	1	5	10.24	2.60	0.61	8.72	2.60	0.63	10.24	4.50	0.55
1	1	7	8.72	2.60	0.72	11.38	2.12	0.64	10.62	2.12	0.55
1	1	9	10.33	2.12	0.56	10.24	5.69	0.66	7.19	3.55	0.59
**Total *J* for S, M, F:** 9.227											
2	0	1	10.33	4.02	0.83	11.47	2.35	0.68	8.71	4.74	0.94
2	0	3	11.76	3.31	0.74	9.19	4.26	0.73	9.48	3.07	0.69
2	0	5	9.86	3.31	0.82	9.95	2.84	0.62	9.10	4.02	0.73
2	0	7	10.33	2.84	0.67	9.19	4.02	0.62	7.57	4.26	1.00
2	0	9	8.34	2.36	0.79	9.19	3.55	0.66	9.19	4.50	0.74
**Total *J* for S, M, F:** 11.261											
2	1	1	8.33	2.59	0.80	9.19	5.45	0.79	10.62	2.83	0.47
2	1	3	10.24	3.07	0.54	9.86	3.78	0.54	10.24	3.07	0.54
2	1	5	7.66	3.55	0.90	10.33	3.07	0.47	8.43	4.02	0.58
2	1	7	11.38	2.84	0.44	10.33	2.35	0.42	8.34	5.22	1.14
2	1	9	11.09	3.31	0.42	8.04	2.35	0.52	6.14	4.50	0.92
**Total *J* for S, M, F:** 9.490											

**Table 3 sensors-19-00312-t003:** Tuning results from 30 additional experiments carried out in ROS-Gazebo (statistics).

–	Slow (S)			Medium (M)			Fast (F)		
	$k_{P}$	$k_{D}$	J	$k_{P}$	$k_{D}$	J	$k_{P}$	$k_{D}$	J
mean value	9.221	3.335	0.640	9.230	3.095	0.670	8.689	3.478	0.740
std deviation	1.190	1.100	0.170	1.230	1.050	0.210	1.130	1.090	0.200
min value	7.190	2.120	0.379	7.190	2.120	0.477	6.430	2.360	0.572
max value	11.000	4.980	0.935	10.620	5.210	1.153	10.240	4.980	1.278

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
