# Peer review of "Real-Time Model-Free Minimum-Seeking Autotuning Method for Unmanned Aerial Vehicle Controllers Based on Fibonacci-Search Algorithm"

_sensors, 2019, doi:10.3390/s19020312_

Reviewer 1 Report

The authors have presented (claimed) a method that can give model free optimal controller for UAV system. The followings are my concern:

1> This is a huristic approach; extensive search is required in field. So, if one wants to run the controller in an unstructured environment, it may not work.

2> Model free and optimal control does not go along. One cannot guarantee optimality of the controller if there is  variations in the structure.

3> The method appears non-convex optimization with no cost-function. So, guaranteeing optimality is even more difficult.

4> From control theoretic point of view, there is no stability analysis, which means the design may work may not work.

5> I can understand that the authors are targetting a working model for competetion. However, one has to give theoretical justification of control design. Otherwise, as a control theorist, I cannot be assured by this design.

The overall work may be a good engineering solution, but claims must be theoretically justified. 

Best Wishes!

Author Response

Dear Reviewer,

Thank you for such an in-depth analysis of our text. The comments are very valuable and allowed us to improve the final version of the article, as well as outline the direction of further work on this subject.

Reviewer 2 Report

The proposed method depends on an optimization procedure using Fibonacci-search technique, intending  to find a global solution regarding the parameter tuning. This reviewer agrees with that this method  is novel. However, the paper has some problems to be solved. Even the paper has many issues, I suggest to publish it after revision.

The authors should consider the following points.

1) In Page 3, the authors discuss Fuzzy  and neural network methods. However, the authors did not give the reference regarding the neural network method.
Please cite the following neural network control papers.
[1]SN Huang, Kok Kiong Tan, Tong Heng Lee,Adaptive friction compensation using neural network approximations,IEEE Transactions on Systems, Man, and Cybernetics, Part C (Applications and Reviews),
vol.30, no.4, pp551-557,2000
[2]SN Huang, Kok Kiong Tan, Tong Heng Lee,Adaptive motion control using neural network approximations, Automatica, vol.38,no.2,pp227-233,2002.

2) In Page 6, Define F and unimodal function in mathematical forms.
In page 6, the authors claim that the optimization should be initialized from different starting points to obtain  a global solution. I am confusing with this point. Since the authors use an on-line tuning method to obtain the PID parameters, how to start from different initial points will be
challenging.  The authors should explain this statement.

3) In [39], it is based on the second-order plus delay model. The method in [39] is not model-free one.

4) See Page 10, Proposition 4.5. the authors rewquire to give an allowable ranges of P1 and P2 ensuring stability of the closed-loop system. This requires to give a theoretical analysis of the closed-loop stability, i.e., we need to give a stability analysis and  find the range of P1 and P2. Unfortunately, the closed-loop stability is based on the UAV model, possible nominal model with uncertain part.

5) In Page 12, the authors should note that when you use (12) (13) to assess the performance, this implies that you depend on the model since (12)(13) are based on the UAV model. If so, the authors should remove the model-free...in the paper.
In Page 17, the authors should give the cost function matematically so that reader can understand your method clearly.

6) In the simulation section, the authors should explain your method clearly and consider the following points:
i)define the parameters range ( how to get these range)
ii)define the cost function mathematically.
iii) Give the 1th -3rd iteration calculation processes.
iv) Give the result.

Author Response

Dear Reviewer,

Thank you for such an in-depth analysis of our text. The comments are very valuable and allowed us to improve the final version of the article, as well as outline the direction of further work on this subject. Please see the attachment.

Round  2

Reviewer 1 Report

Though I agree with most of the responses, I cannot agree on a paper that claims 'optimal control' without any feasiblity/stability study (response 4). I am not asking you to do stochastic analysis. If one is worried about external wind force, then take it as bounded external disturbance and do stability analysis (this is done quite often). This gives you an idea that what should be the choice of gains for a certain wind speed. Or, you can say that given actuator limitations, for this choice of gains these amount of gust can be tolerated. Stability analysis is not a fancy thing to ignore. It gives a safe operting region theoretically what you are 'assuming'. If one is claiming to contribute in 'optimal control' then, at least, I cannot agree without stability analysis. Otherwise, the authors can move with 'optimality of function' rather 'optimal control'.

Further, with only stability analysis, one can really understand whether it is really 'model free' or not. For example, PID control does not require model. Why one should use your method and not PID?

Further, if the authors are worried about stochastic nature in practice, can they prove optimality in stochastic environment with this law? 

Another suggestion: look for works of Prof. F.L Lewis or Vrabi for adaptive-optimal control. There is a complete different branch where people are toiling to achieve something what (stability analysis) you are saying not required.

Author Response

We fully appreciate the comments and bear a strong belief that the current version of the paper makes it more suitable for publication in Sensors. 

Reviewer 2 Report

The authors int this paper present  an optimization procedure using Fibonaccisearch
technique, intending to find a global solution regarding the parameter
tuning. The proposed method is novel, even it has some theoretical issues.  The authors have answered my questions in the last review. But I am not satisfactory with the answer in the point 4.  I attached here again.

4)See Page 10, Proposition 4.5. the authors require to give an allowable
ranges of P1 and P2 ensuring stability of the closed-loop system. This
requires to give a theoretical analysis of the closed-loop stability, i.e.,
we need to give a stability analysis and find the range of P1 and P2.
Unfortunately, the closed-loop stability is based on the UAV model,
possible nominal model with uncertain part.

I can understand that for safety reason you need to have a tuning range. However it is quite important to give a reasonable way to find this range.  I suggest to use an offline training to obtain a rough parameter range ( this needs to do a lot of testing). Please note that no body will use auto-tuning without any knowledge regarding the plant control.  Therefore, the offline tests are necessary.

Author Response

We fully appreciate the comments and bear a strong belief that the current version of the paper makes it more suitable for publication in Sensors. 

Round  3

Reviewer 1 Report

I have no further comments particularly on the technical side. But PID is actually model-free in the sense that it treats the system as a black box. The control input only takes error as the feedback, nothing else. The gains are tuned depending upon the response of the system; but truely, it does not have sound stability analysis and rule for gain selection (more of heuristic/ trial-error approach from personal point of view).

Anyways, guaranteeing optimality is not a big issue from practical point of view as in general, nonlinearities and lack of precise model knowledge are issues.